



# 1 The drivers and health risks of the unexpected surface ozone
# 2 enhancements over the Sichuan basin, China in 2020

Youwen Sun[1, 2], Hao Yin[1, 2,†], Xiao Lu[3,†], Justus Notholt[4], Mathias Palm[4], Cheng Liu[2], Yuan Tian[5],
and Bo Zheng[6]
[1]*Key Laboratory of Environmental Optics and Technology, Anhui Institute of Optics and Fine*
*Mechanics, HFIPS, Chinese Academy of Sciences, Hefei 230031, China*
[2]*Key Laboratory of Precision Scientific Instrumentation of Anhui Higher Education Institutes,*
*University of Science and Technology of China, Hefei, 230026, China*
[3]*School of Atmospheric Sciences, Sun Yat-sen University, Zhuhai, 519082, China*
[4]*University of Bremen, Institute of Environmental Physics, P. O. Box 330440, 28334 Bremen,*
*Germany*
[5]*Institutes of Physical Science and Information Technology, Anhui University, Hefei 230601,*
*China*
[6]*Institute of Environment and Ecology, Tsinghua Shenzhen International Graduate School,*
*Tsinghua University, Shenzhen 518055, China*
[†]Corresponding authors.
E-mail addresses: Hao Yin (yhyh95@mail.ustc.edu.cn) and Xiao Lu (luxiao25@mail.sysu.edu.cn)
**Abstract**
After a continuous increase in surface ozone ($O_3$) level from 2013 to 2019, the overall
summertime $O_3$ concentration across China showed a significant reduction in 2020. In contrast to
this overall reduction in surface $O_3$ across China, unexpected surface $O_3$ enhancements of 10.2 ±
0.8 ppbv (23.4%) were observed in May-June 2020 vs. 2019 over the Sichuan basin (SCB), China.
In this study, we use high resolution nested-grid GEOS-Chem simulation, the eXtreme Gradient
Boosting (XGBoost) machine learning method and the exposure−response relationship to determine
the drivers and evaluate the health risks of the unexpected surface $O_3$ enhancements. We first use
the XGBoost machine learning method to correct the GEOS-Chem model-to-measurement $O_3$
discrepancy over the SCB. The relative contributions of meteorology and anthropogenic emissions
changes to the unexpected surface $O_3$ enhancements are then quantified with the combination of
GEOS-Chem and XGBoost models. In order to assess the health risks caused by the unexpected $O_3$
enhancements over the SCB, total premature death mortalities are estimated. The results show that
changes in anthropogenic emissions caused 0.9 ± 0.1 ppbv of $O_3$ reduction and changes in
meteorology caused 11.1 ± 0.7 ppbv of $O_3$ increase in May-June 2020 vs. 2019. The meteorology-
induced surface $O_3$ increase is mainly attributed to significant increases in temperature and
downward potential vorticity, and decreases in precipitation, specific humidity and cloud fractions
over the SCB and surrounding regions in May-June 2020 vs. 2019. These changes in meteorology
combined with the complex basin effect enhance downward transport of $O_3$ from upper troposphere,
enhance biogenic emissions of volatile organic compounds (VOCs) and nitrogen oxides ($NO_x$),
speed up $O_3$ chemical production, and inhabit the ventilation of $O_3$ and its precursors, and therefore
account for the surface $O_3$ enhancements over the SCB. The total premature mortality due to the
unexpected surface $O_3$ enhancements over the SCB has increased by 89.8% in May-June 2020 vs.
41 2019.



Keywords: Ozone; Health risk; Emissions; Meteorology; Chemical model; Machine learning
**1. Introduction**
Surface ozone ($O_3$) is largely generated from its local anthropogenic (fossil fuel and biofuel
combustions) and natural (biomass burning (BB), lightning, and biogenic emissions) precursors
such as volatile organic compounds (VOCs), nitrogen oxides ($NO_x$), and carbon monoxide (CO) via
a chain of photochemical reactions (Cooper 2019; Sun et al., 2018). Additional portion of surface
$O_3$ is transported from far away regions or from stratosphere (Wang et al., 2020; Akimoto et al.,
2015). Surface $O_3$ is one of the most harmful air pollutants that threatens human health and corps
production (Van Dingenen et al., 2019; Lu et al., 2020; Sun et al., 2018; Fleming et al., 2018).
Exposure to ambient $O_3$ pollution evokes a series of health risks including stroke, respiratory disease
(RD), hypertension, cardiovascular disease (CVD), and chronic obstructive pulmonary disease
(COPD) (Brauer et al., 2016; Liu et al., 2018; Lelieveld et al., 2013; Li et al., 2015; Wang et al.,
2020; Lu et al., 2020). Lu et al. (2020) estimated that the premature RD mortalities attributable to
ambient $O_3$ exposure in 69 Chinese cities in 2019 reached up to 64,370.
Surface $O_3$ variability is sensitive to both emissions and meteorological changes (Liu et al.,
2020a; Liu et al., 2020b; Lu et al., 2019a). Meteorological conditions affect surface $O_3$ variability
indirectly through changes in natural emissions of its precursors or directly via changes in wet and
dry removal, dilution, chemical reaction rates, and transport flux (Lu et al., 2019b; Li et al., 2019a;
Lin et al., 2008; Liu et al., 2020a). A reduction in temperature can lessen $O_3$ production by slowing
down the chemical reaction rates (Lee et al., 2014; Fu et al., 2015; Liu et al., 2020a) or reducing the
biogenic VOCs and $NO_x$ emissions (Guenther et al., 2006; Tarvainen et al., 2005; Im et al., 2011).
A dryer meteorological condition can result in an increase in surface $O_3$ level (Kalabokas et al.,
2015; He et al., 2017; Liu et al., 2020a). Depending which process dominates the influence of
planetary boundary layer height (PBLH) on surface pollutants, a higher PBLH can either reduce
surface $O_3$ level by diluting $O_3$ and its precursors into a larger volume of air (Sanchez-Ccoyllo et al.,
2006; Wang et al., 2020) or increase in surface $O_3$ level by transporting more $O_3$ from upper
troposphere or lessening NO abundance for $O_3$ titration (Sun et al., 2010; He et al., 2017; Liu et al.,
2020a). Transport of $O_3$ from stratosphere to troposphere by synoptic scale and mesoscale process,
as indicated by an increase in potential vorticity (PV), typically leads to surface $O_3$ enhancement
(Wang et al., 2019; Wang et al., 2020). Precipitation has been verified to decrease surface $O_3$ level
through the wet removal of its precursors, and clouds reduce surface $O_3$ level by decreasing the
oxidative capacity of the atmosphere and enhancing scavenging of atmospheric oxidants (Lelieveld
and Crutzen, 1990; Liu et al., 2020b; Seinfeld and Pandis, 2016; Shan et al., 2008). A higher wind
speed can decrease surface $O_3$ level by a fast ventilation of $O_3$ and its precursors (Lu et al., 2019a;
Sanchez-Ccoyllo et al., 2006).
Emissions of air pollutants affect surface $O_3$ variability by perturbing the abundances of
hydroperoxyl ($HO_2$) and alkylperoxyl ($RO_2$) radicals which are the key atmospheric constituents in
formation of $O_3$ (Liu et al., 2020b). Many previous studies have verified a nonlinear relationship
between $O_3$ and its precursors (e.g., Atkinson, 2000; Wang et al., 2017; Liu et al., 2020b; Sun et al.,
2018; Lu et al., 2019). If surface $O_3$ formation regime lies within the VOCs limited region,
reductions in VOCs emissions will result in a reduction in surface $O_3$ level. Similarly, if surface $O_3$
formation regime lies within the $NO_x$ limited region, reductions in $NO_x$ emissions will result in a





reduction in surface $O_3$ level (Atkinson, 2000; Wang et al., 2017). If surface $O_3$ formation regime
lies within transitional region, reductions in either VOC or $NO_x$ emissions will result in a reduction
in surface $O_3$ level. Atmospheric aerosols can affect surface $O_3$ level through either heterogeneous
reactions of reactive gases (Lu et al., 2012; Li et al., 2018; Stadtler et al., 2018; Lou et al., 2014) or
affecting the solar flux for gases photolysis and oxidation (Li et al., 2011; Xing et al., 2017).

6       Understanding the drivers of surface $O_3$ variability has a strong implication for $O_3$ mitigation

purpose (Sun et al., 2018; Lu et al., 2019a). China has experienced a continuous increase in surface
$O_3$ level despite the implementation of control measures on $NO_x$ since 2013 (Liu et al., 2020a, 2020b;
Lu et al., 2018, 2020). Many studies have attempted to determine the drivers of high-$O_3$ events
occurred in specific region and time across China. Most of these studies focus on the most densely
populated and highly industrialized areas in eastern China, whereas the studies in the rest part of
China are still limited (Liu et al., 2020a; Liu et al., 2020b; Lu et al., 2018; Sun et al., 2018; Wang et
al., 2017). As China has a vast territory with a wide range of emission levels and meteorological
conditions, $O_3$ variability and its drivers may vary both temporally and geographically, so the results
from one region are not likely to be applicable nationally. In addition, previous studies typically use
state-of-the-art chemical transport models (CTMs) with sensitivity simulations to quantify the
drivers of $O_3$ variability, e.g., fixed meteorology but varied emission levels to quantify the influences
of emission changes or vice versa (Lu et al., 2019a; Liu et al., 2020a; Liu et al., 2020b). However,
uncertainties in local meteorological fields, emission estimates, and model mechanism can lead to
discrepancy in CTMs that may affect the accuracy of $O_3$ predictions and their sensitivities to changes
in emissions and meteorology (Lu et al., 2019a; Young et al., 2018). This is in particular for the
Sichuan basin (SCB), one of the most industrialized and populated cities cluster in western China,
where large discrepancies between measured and modelled surface $O_3$ are found due to the complex
terrain (Lu et al, 2019a; Wang et al, 2020).
After a continuous increase in surface $O_3$ level from 2013 to 2019, the summertime (May-
August) $O_3$ concentration across China showed a significant reduction in 2020 (Figure 1 (d)). In this
study, we use high resolution nested-grid GEOS-Chem simulation, the eXtreme Gradient Boosting
(XGBoost) machine learning method and the exposure−response relationship to determine the
drivers and evaluate the health risks of the unexpected surface $O_3$ enhancements. We first use the
XGBoost machine learning method to correct the GEOS-Chem model-to-measurement $O_3$
discrepancy over the SCB. The relative contributions of meteorology and anthropogenic emissions
changes to the unexpected surface $O_3$ enhancements are then quantified with the combination of
GEOS-Chem and XGBoost models. In order to assess the health risks caused by the unexpected $O_3$
enhancements over the SCB, total premature death mortalities are also estimated.
**2. Methods and data**
**2.1 Surface $O_3$ data and auxiliary data over the SCB**
China has identified nine cities clusters that lead the populations and developments of economy,
society, and culture across China. The SCB contains the fourth largest cities cluster in China after
the Yangtze River Delta (YRD), the Pearl River Delta (PRD), and the Beijing-Tianjin-Hebei (BTH)
cities clusters. The location of the SCB cities cluster is shown in Figure S1. With Chongqing and
Chengdu as the dual cores, more than a dozen cities including Mianyang, Deyang, Yibin, Nanchong,
Dazhou, and Luzhou over the SCB have achieved rapid economic development and industrial



upgrading. As the region with the strongest economic strength and best economic foundation in
western China, the SCB region has many industries such as energy and chemical industry, electronic
information, food processing, equipment manufacturing, eco-tourism, and modern finance. As one
of the most densely populated and highly industrialized region in China combined with the basin
terrain which is easy to trap air pollutants, the SCB is a newly emerging severe air pollution region
in China.

7       Surface $O_3$ measurements over the SCB are available from the China National Environmental
Monitoring Center (CNEMC) network (http://www.cnemc.cn/en/, last access: 2 July 2021). The
CNEMC network has routinely measured the concentrations of CO, $O_3$, $NO_2$, $SO_2$, $PM_{10}$, and $PM_{2.5}$
at 122 sites in 22 key cities over the SCB since 2015. The mean geolocation, population, the number
of measurement site, data range of each city are summarized in Table 1. The altitude of these cities
ranges from 0.3 to 4.3 km (above sea level, a.s.l.) and the population ranges from 822 to 32,054
thousand. The number of measurement site in each city ranges from 2 to 21. Surface $O_3$
measurements at all measurement sites are based on similar differential absorption ultraviolet (UV)
analyzers. The hourly mean time series of surface $O_3$ concentrations have covered the period from
January 2015 to present at all measurement sites in the 22 cities. After removing unreliable
measurements with the filter criteria used in Lu et al. (2020), we average the $O_3$ concentrations at
all measurement sites in each city to form a city representative $O_3$ dataset. $O_3$ metric used in this
study is on maximum 8-h average (MDA8) basis.
Since the vertical distributions of tropospheric HCHO and $NO_2$ are weighted heavily toward
the lower troposphere over polluted regions, many studies have used tropospheric column
measurements of these gases to represent near-ground conditions (Streets et al., 2013; Sun et al.,
2018; Sun et al., 2021). In this study, the tropospheric $NO_2$ and HCHO columns used for
investigating the changes in $O_3$ precursors in May-June 2020 vs. 2019 are prescribed from the
TROPOMI Level 3 products. TROPOMI overpasses China at approximately 13:30 local time (LT)
with a ground pixel size of 7 km × 7 km. Pixels with quality assurance values of less than 50% for
HCHO and 75% for $NO_2$ are not included in present work.
**2.2 GEOS-Chem nested-grid simulation**
We use the high resolution nested-grid GEOS-Chem model version 12.2.1 to simulate surface
$O_3$ over the SCB (Bey et al., 2001). Simulations are conducted at a horizontal resolution of 0.25 °×
0.3125 ° over the nested domain (70 °-140 °E, 15 °-55 °N) covering China and surrounding regions.
The boundary conditions for the nested-grid GEOS-Chem simulation are archived from the global
simulation at 2 °× 2.5 ° resolution (Yin et al., 2019; Yin et al., 2020; Sun et al., 2021). We spun up
the model for one year to remove the influence of the initial conditions. We first run global
simulation at 2 °× 2.5 ° resolution and then interpolate the restart file on 1 January 2018 into high
resolution (0.25 °× 0.3125 °) for the nested domain to initialize the nested model simulation from
January 2019 to June 2020.
The simulations were driven by GEOS-FP meteorological field at the native resolution of 0.25°
× 0.3125° and 47 layers from surface to 0.01 hPa at the top. The PBLH and surface meteorological
variables are implemented in 1-hour interval and other meteorological variables are in 3-hour
interval. The time step applied in the model for transport is 5 minutes and for chemistry and
emissions is 10 minute (Lu et al., 2019; Philip et al., 2016). The non-local scheme for the boundary





layer mixing process is from Lin et al. (2010), wet deposition is from Liu et al. (2001), and dry
deposition is generated with the resistance-in-series algorithm (Wesely, 1989; Zhang et al., 2001).
The photolysis rates are from the FAST-JX v7.0 photolysis scheme (Bian and Prather, 2002).
Chemical mechanism follows the universal tropospheric-stratospheric Chemistry (UCX)
mechanism (Eastham et al., 2014). The GEOS-Chem simulation outputs 47 layers of $O_3$ and other
atmospheric constituents over China with a temporal resolution of 1 hour.
7         We use the Community Emissions Data System (CEDS) inventory for global anthropogenic
emissions at the latest 2017 level, which is overwritten by the Chinese anthropogenic emissions
with the Multi-resolution Emission Inventory (MEIC) in 2019 (Li et al., 2017; Hoesly et al., 2018;
Zheng et al., 2018). Anthropogenic emissions are fixed for 2019 and 2020. Global BB and biogenic
emissions were from the Global Fire Emissions Database (GFED v4) inventory (Giglio et al. 2013)
and the Model of Emissions of Gases and Aerosols from Nature (MEGAN version 2.1) inventory
(Guenther et al. 2012), respectively. Natural emissions of BB, biogenic VOCs, lightning $NO_x$, and
soil $NO_x$ are calculated online in the model.
**2.3 Correction of GEOS-Chem discrepancy with machine learning method**

16        We used the XGBoost machine learning method to correct the GEOS-Chem model-to-
measurement $O_3$ discrepancy over the SCB. It uses the Gradient Boosting Decision Tree (GBDT)
framework to iteratively train the GEOS-Chem model-to-measurement discrepancy to improve the
model predictions in a stagewise manner. XGBoost method minimizes the loss function by adding
a weak learner for the purpose of optimizing the objective function. The optimization objective
function used in XGBoost model is expressed as,

$$L^{(t)} \simeq \sum_{i=1}^{n} [l(y_i, \hat{y}^{(t-1)}) + g_i f_t(x_i) + \frac{1}{2} h_i f_t^2(x_i)] + \Omega(f_t)$$

$$g_i = \partial_{\hat{y}^{(t-1)}} l(y_i, \hat{y}^{(t-1)}) \quad\quad\quad (1)$$

$$h_i = \partial_{\hat{y}^{(t-1)}}^2 l(y_i, \hat{y}^{(t-1)})$$

22        where $g_i$ and $h_i$ are first and second order gradients of the loss function, respectively. $L^{(t)}$
represents the optimization objective function to be solved at the $t$-th iteration. $l(y_i, \hat{y}^{(t-1)})$ is the
loss function representing the difference between the prediction for the $i$-th sample at the ($t$-1)-th
iteration and the real values $y_i$. Function $f(t)$ is the change amount at the $t$-th iteration. Overall, the
objective function consists of a two order Taylor approximation expansion of the loss function and
the regularization term ($\Omega(f_t)$), which penalize the complexity of the model and prevent overfitting
of the model. Compared to traditional GBDT method, XGBoost method has the following
advantages: (1) effectively handle missing values; (2) prevent overfitting; (3) reduce computing
time by using parallel and distributed computing methods.
31        Since GEOS-Chem model-to-measurement discrepancy is usually site-specific, we train a
separate XGBoost model for each site. Similar to the method of Keller et al. (2021), we use a full
seasonal cycle of hourly measurements in 2019 at each site as the learning samples, and GEOS-
Chem input of emissions and meteorological parameters, output concentrations of atmospheric
constituents, and time information as training input data. In order to incorporate emissions and
meteorological factors that affect $O_3$ production properly, we have included the GEOS-Chem



simulated concentrations of 43 atmospheric chemical constituents, emissions of 21 atmospheric
chemical constituents, 10 meteorological parameters, and 4 time parameters (e.g., hour, day, month,
and year) into the data training. All these training input data are summarized in Table S1 and have
been standardized. We choose a learning rate of 0.35, maximum tree depth of 6, L1 and L2
regularization terms of 0 and 1, the loss function of mean square, and evaluation metric of root-
mean-square error (RMSE) in the data training.

7         We use $k$-fold cross-validation method to test the performance of the XGBoost model ($k=1 -$

$n$). First, all sample data are randomly and uniformly divided into $k$ groups, where one group is
taken as the test dataset and the remaining $k$-1 groups are taken as the training dataset. We then start
to train the model and use the test dataset to evaluate the performance of the trained model. We
repeated this process for $k$ times by using different groups of dataset as the test data. The training
model is finally determined if all the $k$ groups of experiments show similar performances. This
method can obtain a stability and robustness of XGBoost model and avoid overfitting. In this study,
a10-fold cross-validation method is applied, i.e., we divide the $O_3$ measurements in 2019 into 10
groups of sub data: the training dataset accounts for 90% and the test dataset accounts for the
remaining 10% of the total sample data. We also attempted to use 60% and 80% of the sample data
as the training dataset and do not find significant influences on the results, indicating the robustness
of the XGBoost training model.
**2.4 Quantifying meteorological and emissions contributions**

20        We have used the GEOS-Chem only and the combination of GEOS-Chem and XGBoost model

(hereafter GEOS-Chem-XGBoost) to quantify the contributions of meteorology and anthropogenic
emissions to the unexpected surface $O_3$ enhancements over the SCB in 2020. For the GEOS-Chem
method, since the anthropogenic emissions are fixed, the simulated $O_3$ differences between 2020
and 2019 can be attributed to changes in meteorological conditions, which is calculated as,
$$G\_Met = G_{2020} - G_{2019} \qquad (2)$$
The contribution of anthropogenic emissions changes can then be quantified as,
$$G\_Emis = (Meas_{2020} - Meas_{2019}) - G\_Met \qquad (3)$$
where $G\_Met$ and $G\_Emis$ represent the meteorology and anthropogenic emissions contributions,
respectively. $Meas_{2019}$ and $Meas_{2020}$ represent $O_3$ measurements in 2019 and 2020, respectively.
$G_{2019}$ and $G_{2020}$ represent GEOS-Chem $O_3$ simulations in 2019 and 2020, respectively.

31        Since the GEOS-Chem-XGBoost model has corrected the GEOS-Chem model-to-

measurement discrepancy, we assume it can provide accurate predictions to the surface $O_3$
measurements. For predicting $O_3$ evolutions in 2020, all input parameters except anthropogenic
emissions fed into each GEOS-Chem-XGBoost model are updated to match the measurements in
2020, but anthropogenic emissions are fixed at the 2019 levels. As a result, the differences between
the GEOS-Chem-XGBoost predictions for 2020 and the 2020 measurements are attributed to the
changes in anthropogenic emissions (equation (4)). The meteorology induced contributions are then
obtained as equation (5) by subtracting the anthropogenic emissions induced contributions.
$$XG\_Emis = Meas_{2020} - XG_{2020} \qquad (4)$$
$$XG\_Met = (Meas_{2020} - Meas_{2019}) - XG\_Emis \qquad (5)$$
where the acronyms are similar to those in equations (1) and (2) but for GEOS-Chem-XGBoost
method. By correcting the model-to-measurement discrepancy, GEOS-Chem-XGBoost model is





expected to provide a more accurate $O_3$ sensitivity to changes in both meteorology and
anthropogenic emissions.
**2.5 Health risks evaluation**

4        We have assessed the total premature mortalities including all nonaccidental causes,

hypertension, CVD, RD, COPD, and stroke attributed to ambient $O_3$ exposure in all cities over the
SCB in 2019 and 2020. We first calculated the $O_3$ induced daily premature mortalities based on the
exposure−response relationship described in Cohen et al. (2004), which have been used in many
subsequent studies (Anenberg et al., 2010; Liu et al., 2018; Wang et al., 2021). We then added up
the daily premature mortalities within May-June or the whole year to get the total $O_3$ induced
premature mortalities in the respective periods. The population data used in this study include all
age groups, which may result in higher daily mortalities than expected (Liu et al., 2018; Wang et al.,
2021). According to Cohen et al. (2004), the daily premature mortalities attributable to ambient $O_3$
exposure can be estimated by the following equation (Cohen et al., 2004),

$$\Delta x = \begin{cases} 0, & (if \quad C_{meas} - C_{thres} \leq 0) \\ C_{meas} - C_{thres}, & (if \quad C_{meas} - C_{thres} \geq 0) \end{cases} \quad (6)$$

$$\Delta M = y_0[1 - \exp(-\beta\Delta x)] \times Pop \quad (7)$$

14        where $\Delta M$ represents the daily premature mortalities due to ambient $O_3$ exposure. The city

representative daily mean $O_3$ concentration $C_{meas}$ is an average of all measurements in each city.
Variable $y_0$ is the daily baseline mortality rate for each disease averaged from all ages and genders.
We follow the method of Wang et al. (2021) and use the daily $y_0$ value for each disease from the
latest China Health Statistical Yearbook in 2018. β represents the increase in daily mortality as a result
of each 10 μg/cm$^3$ (~ 5.1 ppbv) increase in daily $O_3$ concentration, which is often referred to as the
concentration response function (CRF) in previous studies. We collected the CRF values straightly
from those used in Yin et al. (2017) and Wang et al. (2021). $\Delta x$ represents the incremental $O_3$
concentration relative to the threshold concentration $C_{thres}$ of 35.1 ppbv, which are used following
Lim et al. (2012) and Liu et al. (2018). Pop represents the population exposed in the ambient $O_3$
pollution, which are available from the seventh nationwide population census in 2020 provided by
National Bureau of Statistics of China. The daily $y_0$ and β values for all non-accidental causes,
hypertension, CVD, RD, COPD, and stroke are summarized in Table S2.
**3 Unexpected surface $O_3$ enhancements over the SCB in 2020**

28        Figures 1(a)-(b) show the May-June mean MDA8 $O_3$ concentrations at all measurement sites

over the SCB in 2019 and 2020. The May-June mean MDA8 $O_3$ concentrations averaged over all
cities in the SCB region in 2019 and 2020 are 48.1 ppbv and 58.3 ppbv, which are 11.0 ppbv lower
and 1.2 ppbv higher than those averaged over all Chinese cities in the same period, respectively. As
the most densely populated and highly industrialized region in western China, the land use,
industrialization, infrastructure construction, and urbanization over the SCB have expanded rapidly
in recent years, resulting in the highest anthropogenic emissions of $O_3$ precursors and highest surface
$O_3$ levels in the region (Figure S2). Although the $O_3$ levels in the SCB cities cluster are lower than
those in the three most developed city clusters in eastern China, i.e., the BTH, the Fenwei Plain
(FWP), and the YRD city clusters, the SCB region has been classified by the MEE as a newly



pollution region for $O_3$ mitigation (Sun et al., 2021). Situated in the basin with stationary
meteorological fields combined with high anthropogenic emissions, the SCB cities cluster is
potential to become a new region with frequent high-$O_3$ events after BTH, FWP, and YRD.
We find significant $O_3$ enhancements by $10.2 \pm 0.8$ ppbv (23.4%) (mean $\pm 1\sigma$ standard deviation)
averaged over all cities in the SCB in May-June 2020 vs. 2019 levels (Figure 1(c)). The largest
enhancements are observed in the most densely populated areas around the megacities Chongqing
and Chengdu ($11.8 \pm 0.6$ ppbv (26.0%)). These year-to-year $O_3$ enhancements over the SCB are
record high in the 2015-2020 period, following an increasing change rate of 1.2% $yr^{-1}$ from 2015 to
2017 and then a decreasing change rate of $-0.7\%$ $yr^{-1}$ from 2017 to 2019. These surface $O_3$
enhancements mainly reflect regional emissions and meteorology changes in the SCB and
surrounding regions since the lifetimes of $O_3$ and most of its precursors are too short to undergo
long range transport.
The significant $O_3$ enhancements over the SCB in May-June 2020 vs. 2019 are opposite to the
overall decrease in surface $O_3$ levels across China in the same period (Figure 1 (d)). After a
continuous increase in surface $O_3$ levels from 2013 to 2019 by approximately 5% $yr^{-1}$ (Figure 1(d)),
the MDA8 $O_3$ averaged over all cities outside the SCB across China in May-June 2020 vs. 2019
levels showed a significant reduction of $5.3 \pm 0.5$ ppbv (8.3%). Such $O_3$ reductions are widespread
in the eastern China, especially in the BTH, FWP, and YRD regions.
**4 Model performance assessment**
We use the metrics of normalized root-mean-square error (NRMSE), normalized mean bias
(NMB), and Pearson correlation coefficient ($R$) to assess the performance of the GEOS-Chem-
XGBoost model. For each measurement site, we analysed these metrics for both training (blue) and
test (red) datasets as shown in Figure S3. We define the NRMSE as the RMSE normalized by the
difference between the 95th and 5th percentile concentrations, and NMB as the mean bias
normalized by average concentration at the given measurement site. The formulas of above metrics
are summarized in Section S1.
The GEOS-Chem-XGBoost model predictions for surface $O_3$ over the SCB show no bias when
evaluated against the training data (NMB=0.01), NRMSEs of less than 0.1, and $R$ between 0.93 –
1.0. Compared to the training data, the performances on the test data show a higher variability, with
an average NMB of –0.04, NRMSE of 0.22, and $R$ of 0.83. We find no significant difference in
prediction performance between clean (less than the $C_{thres}$ defined in section 2.5) and polluted
measurement sites. A number of factors likely contribute to relative poorer statistical results at some
sites such as Ganzizhou, Leshan, and Suining. On the one hand, the training data of these sites may
include certain temporal events that are not easily generalizable, such as unusual emissions activity
(e.g., BB, fireworks, closure of nearby point source) or weather patterns that are not properly
observed, which might be prone to overfitting. In addition, the differences in surface $O_3$ variabilities
between the training data and the test data at these sites are relative larger than other sites, which
can contribute to a relative poorer performance.
We use the SHapely Additive exPlanations (SHAP) approach to understand how the GEOS-
Chem-XGBoost model uses the input variables to make a prediction. The SHAP approach is based
on game-theoretic Shapely values and represents a measure of each predictor's responsibility for a
change in the model prediction (Lundberg et al., 2017). The SHAP values are computed separately





for each individual model prediction, which offer detailed insight into the importance of each input
variable to this prediction while also consider the role of variables interactions (Lundberg et al.,
2020; Keller et al., 2021). Figure 2 shows the SHAP value distribution for all $O_3$ predictors averaged
over all cities in the SCB. The results show that any variables that are expected to be associated with
$O_3$ formation affect model $O_3$ prediction. Generally, the temperatures (at the surface, 2 m height,
and 10 m height) are the most important predictors for the GEOS-Chem model-to-measurement
discrepancy over the SCB, followed by the uncorrected GEOS-Chem simulated $O_3$, reactive
nitrogen (e.g., $NO_2$, Peroxyacetyl nitrate (PAN)), atmospheric oxidants ($O_x$, hydrogen peroxide
($H_2O_2$)), fine aerosol, VOCs (Isoprene, $C_3H_8$), hour of the day, and meteorological variables
including horizontal and vertical wind speeds (u10m, v10m). All of these factors have tight
connections to surface $O_3$ formation over the SCB and it is thus not surprising that the GEOS-Chem
model-to-measurement discrepancies are most sensitive to them (Seinfeld and Pandis, 2016).
We have compared the performances of GEOS-Chem and GEOS-Chem-XGBoost in capturing
the measured surface $O_3$ levels. Figure 3 (a) shows the time series of measured and models predicted
$O_3$ concentrations averaged over all cities in the SCB region. Figure 3 (b) shows histogram of the
differences between the GEOS-Chem-XGBoost predictions and the measurements. The GEOS-
Chem simulations generally capture the daily variability of MDA8 $O_3$ over the SCB, but they show
high MB of 7.8 ppbv (17.5%) and RMSE of 15 ppbv across all measurement sites within the SCB
region. The discrepancy can be mainly attributed to uncertainties in the horizontal transport and
vertical mixing schemes simulated by the GEOS-Chem model at a relatively coarse spatial
resolution compared to the measurements at a single point, and also associated with the errors in
emission estimates, chemical mechanism, and sub-grid-scale local meteorological processes.
Especially errors in high SHAP values of $O_3$ predictors are more likely to result in large model-to-
measurement discrepancy. For example, GEOS-Chem model overestimates the correlations
between surface $O_3$ concentration and temperature (Figure S5 (a)), indicating that this
overestimation of $O_3$-to-temperature sensitivity is one of the major factors contributing to higher
GEOS-Chem model $O_3$ predictions.
By iteratively training and correcting the GEOS-Chem model-to-measurement discrepancy in
$O_3$-to-temperature sensitivity, the correlations between surface $O_3$ concentration and temperature
predicted by the GEOS-Chem-XGBoost model were in good agreement with the measurements
(Figure S5 (a)). With respect to the performance of reproducing the sensitivities of $O_3$ to other
meteorological parameters such as humidity, cloud fraction, and precipitation, the GEOS-Chem-
XGBoost model is also better than the GEOS-Chem (Figure S5 (b)-(d)). After correcting the errors
in all $O_3$ predictors, the GEOS-Chem-XGBoost model significantly improves the prediction of
surface $O_3$ concentrations in all cities over the SCB compared to the GEOS-Chem (Figure S6). It
shows a MB of 0.5 ppbv and RMSE of 0.3 ppbv for all $O_3$ measurements in 2019 over the SCB. As
a result, the overall GEOS-Chem-XGBoost model performance is acceptable and can support
further investigation of the drivers of the unexpected surface $O_3$ enhancements over the SCB in
May-June 2020.
**5 Attribution**
**5.1 Separation of meteorological and anthropogenic emissions contributions**
We attribute quantitatively the surface $O_3$ enhancements in May-June 2020 over the SCB to





changes in anthropogenic emissions and meteorological conditions according to equations (3) and
(4). Differences between the measured and GEOS-Chem-XGBoost predicted $O_3$ in May-June 2020
represent the anthropogenic emissions-induced $O_3$ changes in 2020 vs. 2019, as indicated by the
shadings in Figure 4(a). Figure 4(b) summarizes the mean contributions driven by changes in
anthropogenic emissions and meteorological conditions. Due to different change rates in
anthropogenic emissions in May and June in 2020 (see section 5.3), the changes in anthropogenic
emissions caused an overall increase in surface $O_3$ level in May but a reduction in surface $O_3$ level
in June (Figure 4 (a)). For the May-June mean contributions averaged over all cities in the SCB,
changes in anthropogenic emissions caused $0.9 \pm 0.1$ ppbv of $O_3$ reduction and changes in
meteorology caused $11.1 \pm 0.7$ ppbv of $O_3$ increase, which correspond to $-8.0\%$ and $108\%$ of
relative contributions to the total $O_3$ enhancement ($10.2 \pm 0.8$ ppbv) over the SCB in May-June 2020,
respectively. As a result, the unexpected $O_3$ enhancements over the SCB in 2020 were attributed to
that the anthropogenic emissions induced $O_3$ reductions are dominantly overwhelmed by the
meteorology induced $O_3$ increases.
We compare the meteorology and anthropogenic emissions induced contributions to the
unexpected surface $O_3$ enhancements estimated by the GEOS-Chem-XGBoost model with those by
the GEOS-Chem model only (Figure 4 (b)). Both methods agree that changes in meteorology play
a significant role in interpreting the $O_3$ enhancements, while the absolute magnitudes differ slightly
with each other. For example, the anthropogenic emissions induced $O_3$ reduction calculated with
the GEOS-Chem model only is 0.94 ppbv, while the value for GEOS-Chem-XGBoost model is 1.36
ppbv. By taking the subtraction in equation (1) and the average over all cities, the propagation of
systematic model discrepancies that are common to all measurements sites was effectively
minimized, which can mitigate the difference in attribution results between the GEOS-Chem and
GEOS-Chem-XGBoost methods. However, as demonstrated in Figure S6, model discrepancies may
differ from one region to the other and from time to time. Therefore, the GEOS-Chem-XGBoost
approach is expected to provide a more accurate and consistent estimate on $O_3$ change attribution.
**5.2 Meteorological contribution**
Figure 5 shows the terrain elevations and May-June mean wind fields and surface pressures
over the SCB and surrounding regions. The terrain altitudes are at a resolution of $3 \times 3$ arc-minute,
which indicates a rapid change in altitude from the Tibetan Plateau (4.0 –5.0 km) and Yunan-
Kweichou Plateau (2–3 km) to the SCB (0.5 km). The SCB is located in the saddle between the
Tibetan and Yunnan-Kweichou Plateau (Chen et al., 2009; Sun et al., 2021b). Figure 5 (b) are the
May-June mean wind fields at 500 m overlaid with surface pressure available from GEOS-FP fields
at $0.25\,^\circ \times 0.3125\,^\circ$ resolution. In May-June, the western Pacific Subtropical High originated from
the middle region of the Tibetan Plateau shifts westward to the west of the SCB (Chen et al., 2009).
The southwesterly East Asian summer monsoon generates a cyclonic pattern over the southeast part
of the SCB. Driven by large scale circulations, southwesterly flow enters the east part of the SCB
near the northwest edge of the Yunnan-Kweichou Plateau, while strong northwesterly flow enters
the SCB near the east edge of the Tibetan Plateau. The interaction of these two flows results in a
convergent zone of northward jet stream over the east part of the SCB due to the westward shift of
the Western Pacific Subtropical High and the blocking effect of topography. Furthermore, strong
instability of vertical convection could originate over the basin and move toward the east part of the





SCB as dry air continuously entered the upper layer over the SCB (Chen et al., 2009). This process will continuously intensify the cyclonic vorticity over the SCB, and make it a favorable region for stationary low-level vortices, which tend to trap air pollutants within the SCB region and is referred to as the Southwest Vortex (Chen et al., 2009; Liu et al., 2003).

Figure 6 shows the May-June mean differences in potential vorticity (PV), precipitation, temperature, specific humidity, cloud fraction, and PBLH between 2020 and 2019. In May-June 2020, the northwest, central western and southern China experienced anomaly strong or even record-breaking droughts (https://quotsoft.net/air/), leading to significant increases in temperature and decreases in precipitation, specific humidity and cloud fractions compared to the 2019 levels. These meteorological conditions could enhance biogenic VOCs emissions, speed up $O_3$ chemical production, and the aforementioned basin effect inhabit the ventilation of $O_3$ and its precursors, which contributed to the $O_3$ enhancements over the SCB. Although higher PBLH over the SCB in May-June 2020 vs. 2019 could reduce surface $O_3$ levels by diluting $O_3$ and its precursors into a larger volume of air, this reduction effect was overwhelmed by its enhancement effect, i.e., higher PBLH enhanced downward transport of $O_3$ from the upper troposphere. Indeed, we observed an increase in downward potential vorticity (PV) over the SCB in May-June 2020 vs. 2019 (Figure 6 (a)). It is worth noting that, with similar meteorological conditions in May-June 2020 vs. 2019, the $O_3$ enhancements were not observed in the northwest China such as Xinjiang and Inner Mongolia provinces, and southern China such as the Pearl River Delta (PRD) region, which is also one of the nine well-developed city clusters in China with severe air pollution. This can be partly attributed to low anthropogenic emissions of $O_3$ precursors in northwest China (Lu et al. 2019; Zheng et al. 2018); and that strong exchange between the land and sea in the coastal regions driven by the summer monsoon facilitates the ventilation of $O_3$ and its precursors in the PRD region. Furthermore, the meteorology induced $O_3$ enhancements are probably overwhelmed by the anthropogenic emissions induced $O_3$ reductions in the aforementioned two regions.

**5.3 Emissions contribution**

To suppress the spread of coronavirus pandemic 2019 (COVID-19) across China and above, the Chinese government sealed off several cities starting in January 2020; this included closing local businesses and halting public transportation at an unprecedented scale (Steinbrecht et al., 2021; Liu et al., 2020). These prevention measures quickly spread nationwide. Although the COVID-19 lockdowns in all cities have been removed before May, there are still restrictions on public transportation, businesses, social activities and industrial manufactures, which could cause domestic anthropogenic emissions reductions in both HCHO and $NO_x$. Furthermore, the MEE has implemented The 2020 Action Plan on VOCs Mitigations in 2020. This Action Plan issues a number of control measures including implementation of stringent VOCs emission standards, replacement of raw and auxiliary materials with low VOCs content, and mitigation of unorganized emissions. Driven by above two factors, the TROPOMI observed tropospheric HCHO and $NO_2$ over China in May-June 2020 vs. 2019 reduced by 2.0 ± 0.3% (averaged for all Chinese cities) and 1.1 ± 0.2%, respectively. Due to the relative short lifetime of both HCHO and $NO_2$ in troposphere, these reductions mostly reflect local emissions changes. These reductions in domestic anthropogenic emissions dominated the significant reduction of summertime MDA8 $O_3$ across China in 2020 vs 2019.





We have used the HCHO/NO$_2$ ratios following the method of Sun et al. (2018) to investigate
the O$_3$ production regime over the SCB region. The results show that the satellite observations of
HCHO/NO$_2$ ratios in May-June in most cities over the SCB have indicated a shift toward high values
from 2019 to 2020 but the O$_3$ chemical sensitivity in 2020 still lies within the transitional regime
(Jin et al., 2015; Jin et al., 2017; Figure S7). Meanwhile, the O$_3$ chemical sensitivity in May 2020
is similar to that in June, indicating that the O$_3$ variability in May-June 2020 is sensitive to both NO$_x$
and VOCs. The recently available Chinese anthropogenic emissions statistic data provided by the
MEE show that the anthropogenic VOCs over the SCB has decreased by 5.0% and 3.5% in May
and June in 2020 relative to the 2019 level, respectively. The anthropogenic NO$_x$ in the same period
has increased by 1.5% and decreased by 1.7%, respectively (Zheng et al., 2021). The increase in
anthropogenic NO$_x$ in May 2020 vs. 2019 is attributed to an increase in NO$_x$ emission from power
plant sector, which was not affected by the post-lockdown restrictions for suppressing the spread of
COVID-19 (Table S3). For the May-June aggregation, the anthropogenic VOCs and NO$_x$ over the
SCB have decreased by 4.3% and 0.3%, respectively (Zheng et al., 2021). These independent
analyses on anthropogenic emissions explain the different predicted O$_3$ changes due to
anthropogenic emissions alone in May (increase) versus June (decrease) in the SCB.

17       In contrast to the widespread reductions in both HCHO and NO$_2$ across the BTH, FWP, and
YRD regions, we find notable increases in both HCHO and NO$_2$ in the SCB in May-June 2020 vs.
2019 levels. The tropospheric HCHO and NO$_2$ columns averaged over all cities in the SCB region
have been increased by (2.8 ± 0.3%) and (5.1 ± 0.5%) in 2020 vs. 2019 levels, respectively. Since
both anthropogenic VOCs and NO$_x$ emissions in the SCB showed decreasing change rates in May-
June 2020 vs. 2019, these regional increases in both HCHO and NO$_2$ could thus be attributed to
natural emissions enhancements in both VOCs and NO$_2$ in the SCB. Indeed, natural emissions of
biogenic VOCs and soil NO$_x$ calculated online in the GEOS-Chem model show increasing change
rates in May-June 2020 vs. 2019 in the SCB and surrounding regions (Figure 7). These enhanced
biogenic VOCs and NO$_x$ emissions are most likely driven by the hotter and dryer meteorological
conditions in the SCB and surrounding regions (Figure 7).

28       Finally, we concluded that natural emissions enhancements of both NO$_x$ and VOCs induced by
the unexpected meteorology could be accounted for the O$_3$ enhancements in May-June 2020 over
the SCB. In present work, we were not able to determine which specific VOCs species are the most
effective for O$_3$ enhancements and cannot quantify the relative contributions of VOCs and NO$_x$
enhancements to the O$_3$ enhancements in the SCB. A series of sensitivity studies might be able to
address this important issue, but this is beyond the scope of present work.
**6 Health risks for the O$_3$ enhancements over the SCB**

35       Figure 8 presents the total premature mortalities from all non-accidental causes, hypertension,
CVD, RD, COPD, and stroke attributable to ambient O$_3$ exposure in all cities over the SCB during
May-June in 2019 and 2020. The statistical results for each city in 2019 and 2020 are summarized
in Table S4 and S5, respectively. The surface O$_3$ enhancements over the SCB in May-June 2020 vs.
2019 results in dramatically higher health risks. The estimated total premature mortalities from all
non-accidental causes due to the surface O$_3$ enhancements in May-June 2020 over the SCB is 5455,
which is 89.8% higher than that in the same period in 2019 (i.e., 2874). All above O$_3$ induced
diseases over the SCB have significant increases in total mortalities in May-June 2020 vs. 2019.



The highest health risk among these diseases is from CVD which is 741 in May-June 2019, followed by RD (236), COPD (231), and hypertension (223). This $O_3$ induced health risk rank over the SCB is consistent with those in the YRD, BTH, and PRD in previous studies (Liu et al., 2018; Lu et al., 2020; Yin et al., 2017; Wang et al., 2021). In May-June 2020, total mortalities from CVD, RD, COPD, hypertension, and stroke over the SCB reached up to 1405, 450, 439, 418, and 46, respectively, due to significant $O_3$ enhancements. The change rates for these diseases are 89.6, 90.7, 90.1, 87.4, and 91.7%, respectively.

From a whole year view, the estimated total premature mortalities from all non-accidental causes due to surface $O_3$ exposure over the SCB in 2019 and 2020 are 16,772 and 18,301, respectively (Tables S4 and S5). All $O_3$ induced diseases within May-June 2019 account for about ~ 17.0% of those in the whole year 2019, and this percentage reaches up to ~ 30.0% in 2020 (Figure S8). The total premature mortalities from all non-accidental causes due to surface $O_3$ exposure over the SCB has increased by 1528 in the whole year 2020 vs. 2019 (Figure S9), which is 40.8% lower than that within May-June 2020 vs. 2019 (i.e., 2581). This indicates that the $O_3$ level over the SCB showed an overall decreasing change rate in all months except May-June in 2020 vs. 2019, which resulted in a decrease (by 1053) in $O_3$ induced diseases in the period.

We further investigated the $O_3$ induced diseases in the two most densely populated cities over the SCB (i.e., Chengdu and Chongqing) during May-June in 2019 and 2020. The premature mortalities from all $O_3$ induced diseases in 2020 vs. 2019 in each city are dependent on regional population, surface $O_3$ level, and enhancement level (equation (6)). With largest populations and highest $O_3$ enhancements, the estimated total premature mortalities in Chengdu and Chongqing accounted for 46.9% of total $O_3$ induced mortalities over the SCB during May-June 2020 (Figure 8 (b)-(c)). Since the $O_3$ level and enhancement in Chengdu are larger than those in Chongqing, the total $O_3$ induced mortalities in Chengdu are larger than those in Chongqing, though the population in Chengdu is lower than that in Chongqing. The change rates for all $O_3$ induced diseases in Chengdu are about 75% and in Chongqing are about 160% during May-June 2020 vs. 2019, which are much higher than the enhancement percentages in the two cities (29.9 %). In order to reduce the $O_3$ induced health risk, strident $O_3$ control policies are necessary in densely populated cities.

## 7 Conclusions

Understanding the drivers and health risks of surface high $O_3$ events has a strong implication for $O_3$ mitigation purpose. After a continuous increase in surface $O_3$ level from 2013 to 2019, the overall summertime $O_3$ concentration across China showed a significant reduction in 2020. In contrast to this overall reduction in surface $O_3$ level across China, unexpected surface $O_3$ enhancements of $10.2 \pm 0.8$ ppbv (23%) were observed in May-June 2020 vs. 2019 over the Sichuan basin (SCB), China. In this study, we have used high resolution nested-grid GEOS-Chem simulation, the eXtreme Gradient Boosting (XGBoost) machine learning method and the exposure−response relationship to determine the drivers and evaluated the health risks of the unexpected surface $O_3$ enhancements.

By iteratively training and correcting the GEOS-Chem model-to-measurement discrepancies, the GEOS-Chem-XGBoost model significantly improves the prediction of surface $O_3$ concentrations compared to the GEOS-Chem. It shows a MB of 0.5 ppbv and RMSE of 0.3 ppbv against all $O_3$ measurements over the SCB. As a result, the overall GEOS-Chem-XGBoost model





performance is acceptable and can support further investigation of the drivers of the unexpected surface $O_3$ enhancements over the SCB in May-June 2020. The results show that changes in anthropogenic emissions caused $0.9 \pm 0.1$ ppbv of $O_3$ reduction and changes in meteorology caused $11.1 \pm 0.7$ ppbv of $O_3$ increase. The meteorology-induced surface $O_3$ increase is mainly attributed to significant increases in temperature and downward potential vorticity, and decreases in precipitation, specific humidity and cloud fractions over the SCB and surrounding regions in 2020 vs. 2019 levels. These changes in meteorology combined with the complex basin effect enhance downward transport of $O_3$ from the upper troposphere and biogenic emissions of VOCs and $NO_x$, speed up $O_3$ chemical production, and inhibit the ventilation of $O_3$ and its precursors, and therefore account for the surface $O_3$ enhancements over the SCB.

The unexpected surface $O_3$ enhancements over the SCB in May-June 2020 vs. 2019 result in dramatically higher health risks. The estimated total premature mortalities due to the unexpected surface $O_3$ enhancements over the SCB in May-June 2020 is 5455, which is 89.8% higher than that in the same period in 2019 (i.e., 2874). We further investigated the $O_3$ induced diseases in the two most densely populated cities over the SCB (i.e., Chengdu and Chongqing) during May-June in 2019 and 2020. With largest populations and highest $O_3$ enhancements, the estimated total premature mortalities in Chengdu and Chongqing accounted for 46.9% of total $O_3$ induced mortalities over the SCB. The change rates for all $O_3$ induced diseases in Chengdu are about 75% and in Chongqing are about 160% during May-June 2020 vs. 2019, which are much higher than the enhancement percentages in the two cities (29.9 %). In order to reduce the $O_3$ induced health risks, strident $O_3$ control policies are necessary in densely populated cities.

*Code and data availability.* Surface $O_3$ measurements over the SCB are from http://www.cnemc.cn/en/. All other data are available on request of YS (ywsun@aiofm.ac.cn)

*Author contributions.* YS designed and wrote the paper. HY carried out the GEOS-Chem simulations and GEOS-Chem-XGBoost training and evaluation. XL designed the concept of health risk evaluation and revised the manuscript. BZ constructed the latest MEIC emission inventory. JN, MP, CL, and YT provided constructive comments.

*Competing interests.* None.

*Acknowledgements.* This work is supported by the Youth Innovation Promotion Association, CAS (No.2019434) and the Sino-German Mobility programme (M-0036).

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



# Tables

**Table 1**. Measurement sites in the SCB city clusters. All sites are organised alphabetically. Population statistics are based on the seventh nationwide population census in 2020 provided by National Bureau of Statistics of China.

| Name | Longitude mean | Latitude mean | Altitude mean (km) | Population | Number of sites | Time period |
|---|---|---|---|---|---|---|
| Abazhou | 102.21°E | 31.91°N | 3.5 | 822,587 | 3 | 2015 - present |
| Bazhong | 106.75°E | 31.85°N | 0.8 | 2,712,894 | 4 | 2015 - present |
| Chengdu | 104.04°E | 30.69°N | 0.5 | 20,938,000 | 10 | 2015 - present |
| Chongqing | 106.51°E | 29.58°N | 0.4 | 32,054,200 | 21 | 2015 - present |
| Dazhou | 107.5°E | 31.22°N | 1.0 | 5,385,422 | 5 | 2015 - present |
| Deyang | 104.39°E | 31.12°N | 0.5 | 3,456,161 | 4 | 2015 - present |
| Ganzizhou | 101.96°E | 30.05°N | 3.5 | 1,107,431 | 2 | 2015 - present |
| Guangan | 106.63°E | 30.48°N | 1.7 | 3,254,883 | 6 | 2015 - present |
| Guangyuan | 105.85°E | 32.44°N | 2.1 | 2,305,657 | 4 | 2015 - present |
| Leshan | 103.76°E | 29.57°N | 0.5 | 3,160,168 | 4 | 2015 - present |
| Liangshanzhou | 102.28°E | 27.87°N | 2.3 | 4,858,359 | 5 | 2015 - present |
| Luzhou | 105.43°E | 28.9°N | 0.3 | 4,254,149 | 4 | 2015 - present |
| Meishan | 103.85°E | 30.07°N | 0.8 | 2,955,219 | 6 | 2015 - present |
| Mianyang | 104.73°E | 31.48°N | 0.7 | 4,868,243 | 4 | 2015 - present |
| Nanchong | 106.09°E | 30.8°N | 0.3 | 5,607,565 | 6 | 2015 - present |
| Neijiang | 105.05°E | 29.59°N | 0.5 | 3,140,678 | 4 | 2015 - present |
| Panzhihua | 101.69°E | 26.56°N | 2.6 | 1,212,203 | 5 | 2015 - present |
| Suining | 105.71°E | 30.58°N | 0.5 | 2,814,196 | 4 | 2015 - present |
| Yaan | 103.01°E | 29.99°N | 3.1 | 1,434,603 | 4 | 2015 - present |
| Yibin | 104.62°E | 28.78°N | 2.0 | 4,588,804 | 6 | 2015 - present |
| Zigong | 104.75°E | 29.35°N | 0.3 | 2,489,256 | 6 | 2015 - present |
| Ziyang | 104.64°E | 30.13°N | 0.5 | 2,308,631 | 5 | 2015 - present |

**Figures**

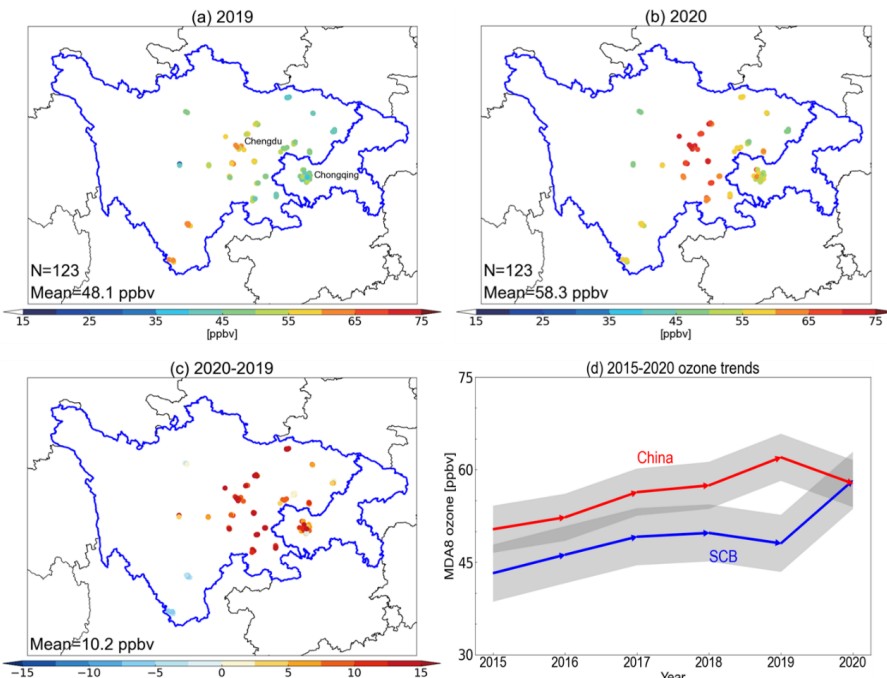

**Figure 1** Surface O$_3$ enhancements over the SCB region in May-June 2020 vs. 2019. (a) Spatial distributions of
May-June mean O$_3$ concentrations over the SCB region in 2019. Number (N) denotes available measurement sites
for this year. We average the O$_3$ concentrations at all measurement sites in each city to form a city representative O$_3$
dataset. (b) Same as (a) but for 2020. (c) Differences between 2020 and 2019. (d) Trends in May-June mean ozone
concentrations from 2015-2020 averaged for all Chinese cities (red) and for the SCB cities cluster (blue). Grey
shadings represent the range of mean value ± 1σ STD across all cities.





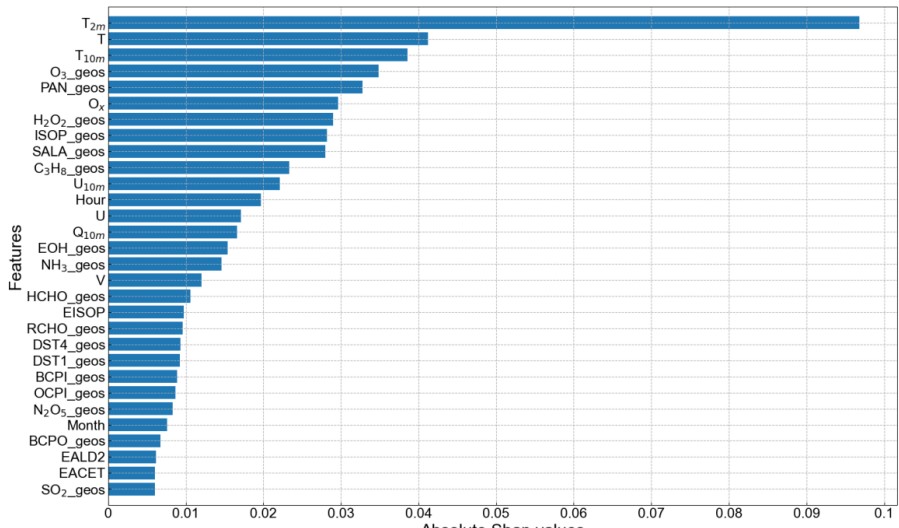

**Figure 2** Importance of input variables for the XGBoost model trained to correct the GEOS-Chem model-to-
measurement $O_3$ discrepancy over the SCB. Shown are the distribution of the SHAP values for each variable
averaged over all cities in the SCB, ranked by the average importance of each feature. Higher SHAP value indicates
higher feature importance. Descriptions for all acronyms are listed in Table A1. For clarity, only the top 30 variables
are shown. See Figure S4 for importance of all variables.





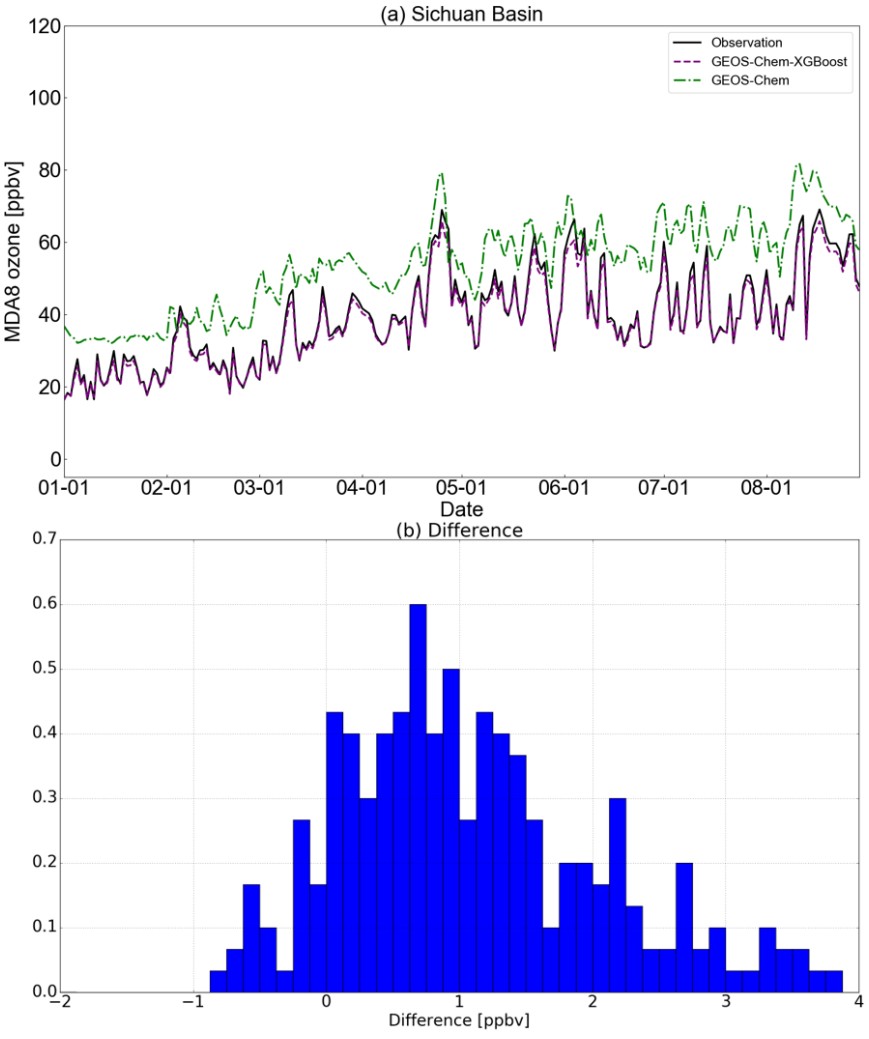

2     **Figure 3** Measured and modelled $O_3$ variabilities over the SCB in 2019 (a). Measured, GEOS-Chem, and GEOS-

3     Chem-XGBoost predicted $O_3$ values are denoted by black solid, grey dashed, and purple dashed lines, respectively.

4     (b) Histogram of the differences between the GEOS-Chem-XGBoost predictions and the measurements.


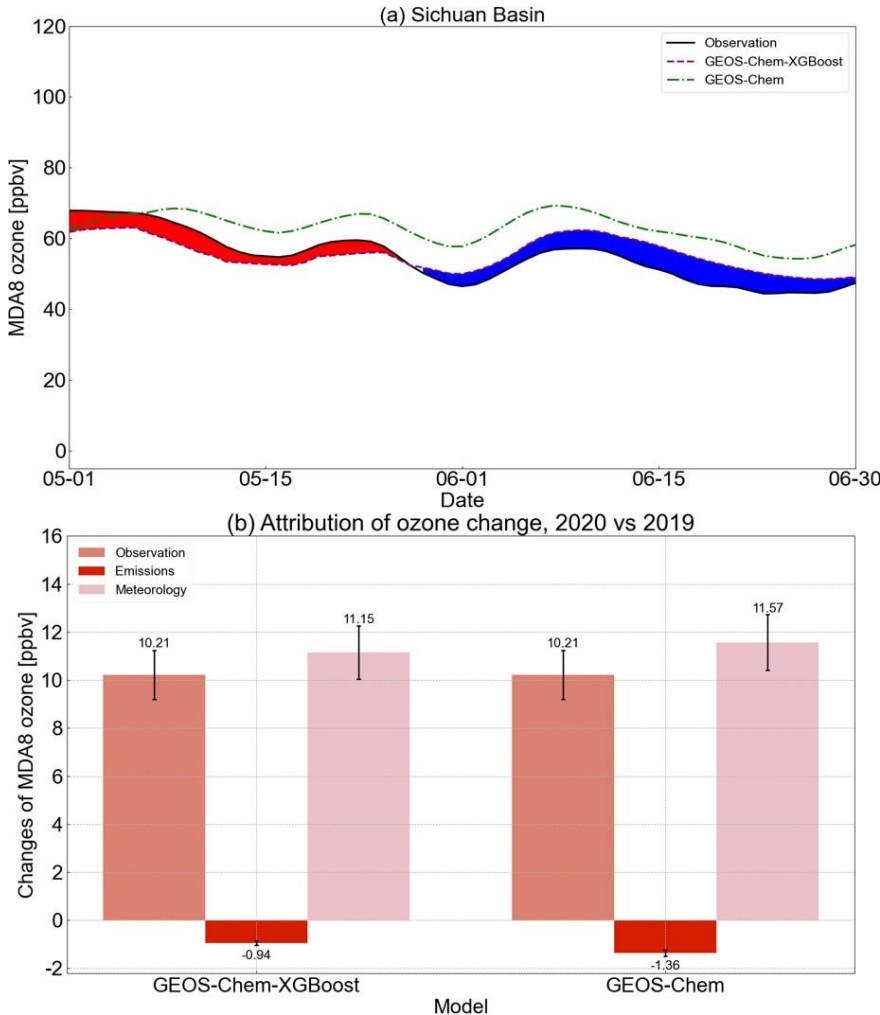

**Figure 4** (a) Comparison of the GEOS-Chem-XGBoost $O_3$ predictions to the 2020 measurements. Red (blue)
shadings represent where GEOS-Chem-XGBoost predictions are higher (lower) than the actual measurements in
2020, indicating that changes in anthropogenic emission lead to $O_3$ increase (decrease) in 2020. (b) Attribution of
surface $O_3$ enhancements over the SCB in May-June 2020 vs. 2019. Filled colored bars denote $O_3$ change as seen
from measurements, and due to changes in anthropogenic emission and meteorological conditions estimated by the
GEOS-Chem-XGBoost model and the GEOS-Chem model. Black vertical bars represent 1σ STD across cities.


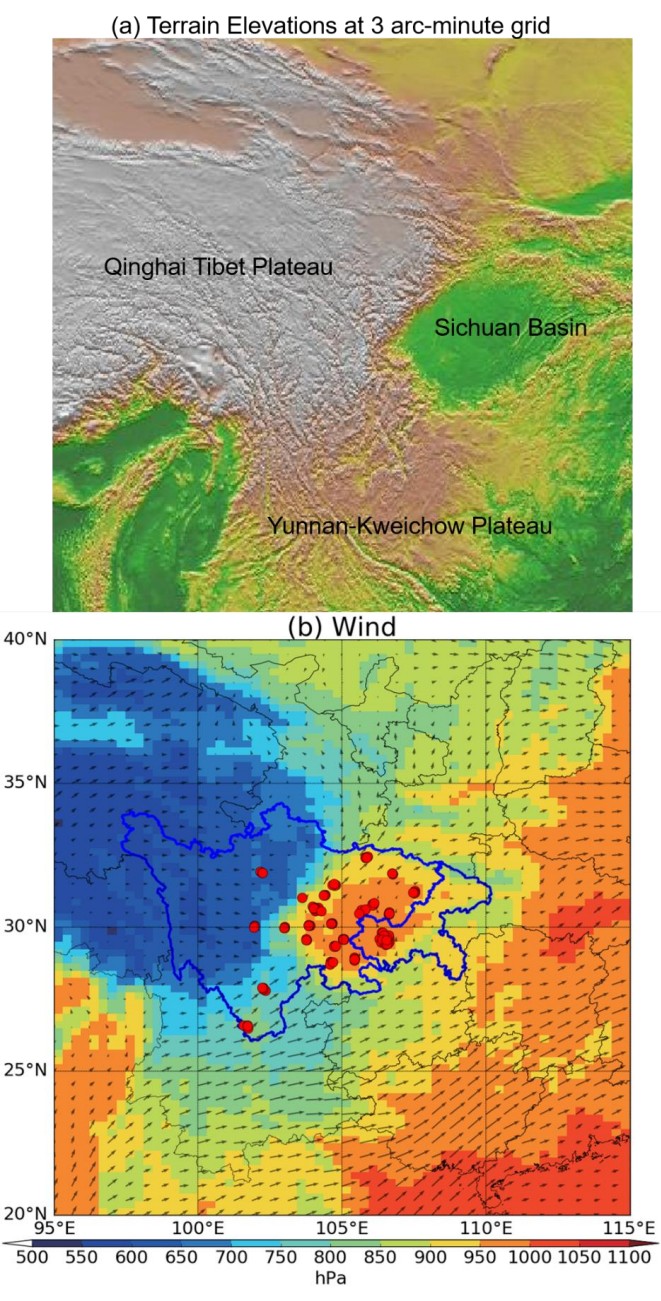

**Figure 5** Terrain elevations (a) and surface temperature and wind fields (b) over the SCB and surrounding regions.
The spatial resolutions for (a) and (b) are 3 × 3 arc-minute and 0.25° × 0.25°, respectively. The white area in black
line is Tibetan Plateau (with altitudes of 4–5 km a.s.l.), the yellow area in red line is the Yunnan-Kweichou Plateau
(2–3 km a.s.l), the green area in circle is the SCB (0.5–1 km a.s.l).

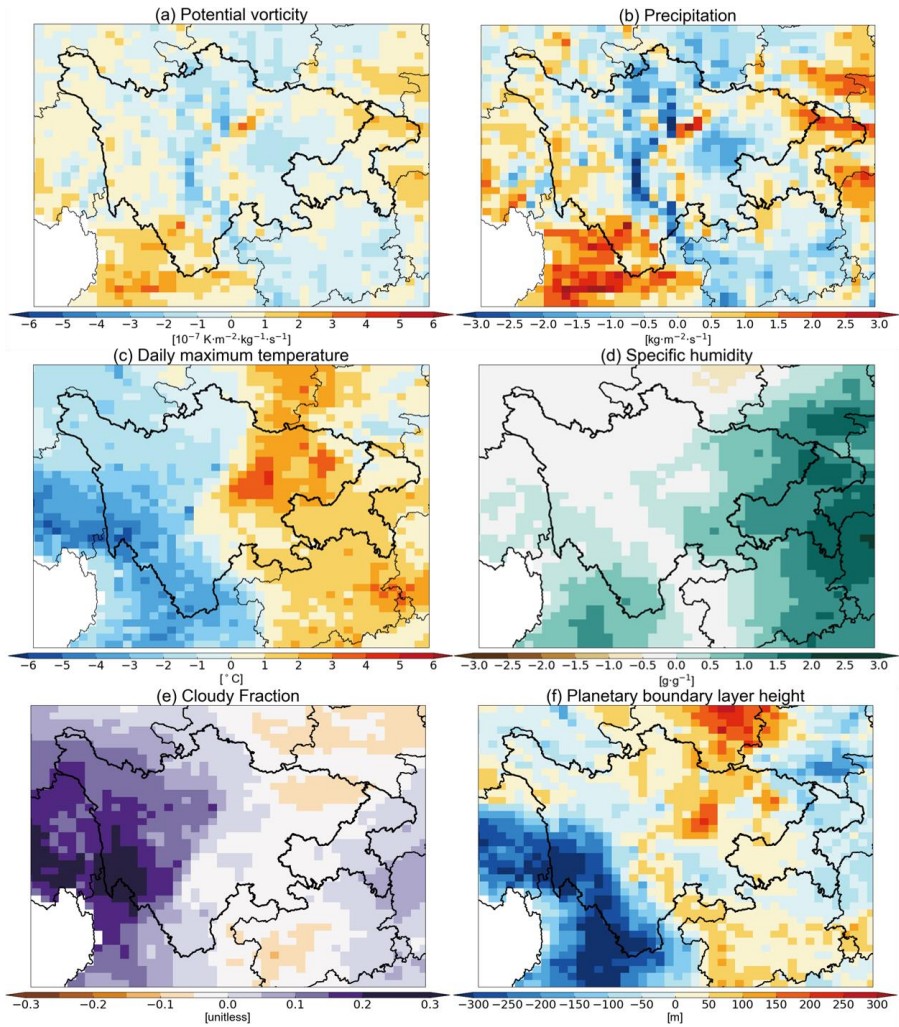

**Figure 6** May-June mean differences in PV (a), precipitation (b), temperature (c), specific humidity (d), cloud

fraction (e), and PBLH (f) between 2020 and 2019 over the SCB and surrounding regions. All these meteorological

parameters are from the GEOS-FP dataset. PV is prescribed at the PBLH and others are at the surface.

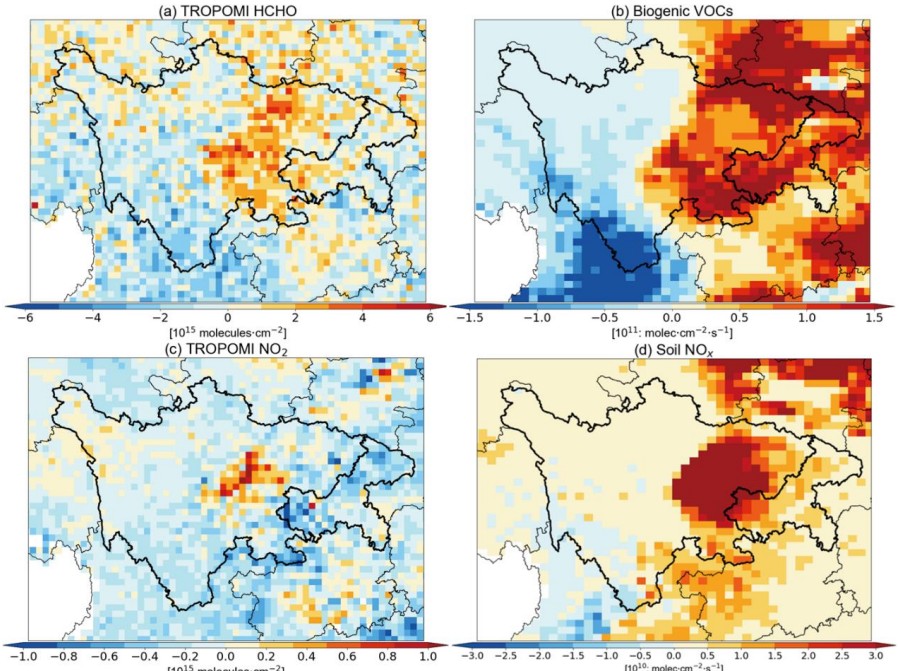

2 **Figure 7** May-June mean differences in $O_3$ precursors between 2020 and 2019. (a) TROPOMI observed HCHO, (b)

3 biogenic VOCs, (c) TROPOMI observed $NO_2$, and (d) Soil $NO_x$. Biogenic VOCs and soil $NO_x$ are available from

4 GEOS-Chem model online calculations.



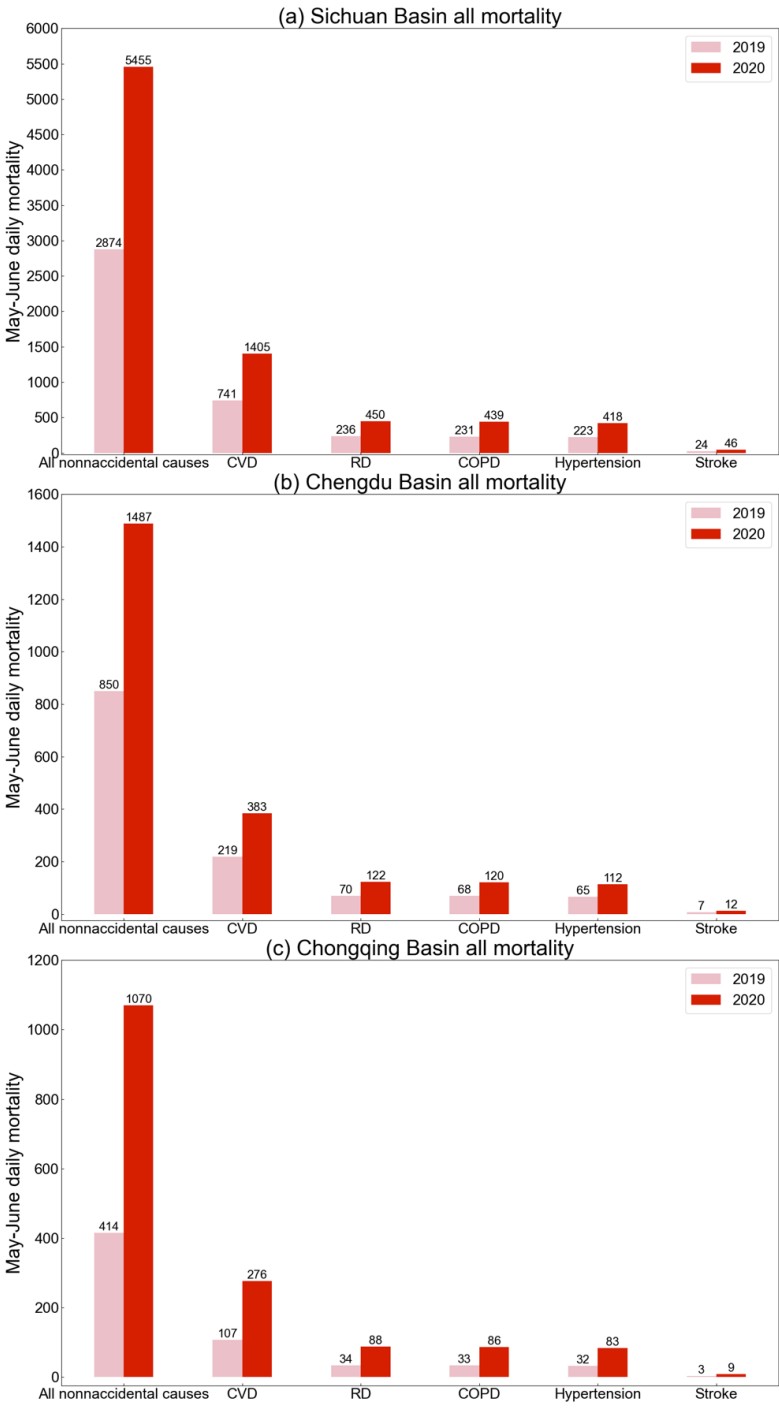

2 **Figure 8** Total daily mortality from all non-accidental causes, CVD, RD, COPD, hypertension, and stroke

3 attributable to ambient $O_3$ exposure over the SCB during May-June in 2019 and 2020.