# Peer review of "The drivers and health risks of the unexpected surface ozone"

_Atmospheric Chemistry and Physics, 2021_

## Author Comment (AC3)

**Supplement of resonse letter #2**

**Section S1**

As the parameters listed in Table S1 of the manuscript are different in units and magnitudes, which could lead to unstable performance of the training model. Therefore, we standardized all the parameters before using them for model training. The standardized process is expressed as below:

$$D_i = \frac{P_i - \mu}{\sigma} \tag{1}$$

where $P_i$, $\mu$, and $\sigma$ are the $i$-th parameter, the average, and the standard deviation of the training input dataset listed in Table S1 of the manuscript, respectively. $D_i$ represents the standardized value used for model training.

[Figure]

**Figure 1.** Probability density functions (PDFs) of hourly planetary boundary layer height (PBLH), temperature at 2 m, and relative humidity in the whole 2019 (blue) and May-June 2020 (red) at Chengdu and Chongqing cities over the SCB, from the GEOS-FP meteorology fields that are used to drive the GEOS-Chem model and to train/predict the ozone bias. We group each data to 10 bins, and frequencies are calculated for each bin.

[Figure]

**Figure 2.** Same as Fig.1, but for hourly concentrations of GEOS-Chem $NO_2$, CO, and HCHO. The difference between 2019 and 2020 in GEOS-Chem only reflects meteorological effects.

[Figure]

**Figure 3** Terrain elevations (a) and surface temperature and wind fields (b) over the SCB and surrounding regions. The spatial resolutions for (a) and (b) are 3 × 3 arc-minute and 0.25° × 0.25°, respectively. The white area in black line is Tibetan Plateau (with altitudes of 4–5 km a.s.l.), the yellow area in red line is the Yunnan-Kweichou Plateau (2–3 km a.s.l), the green area in circle is the SCB (0.5–1 km a.s.l).

[Figure]

**Figure 4** May-June mean differences in vertical pressure velocity (a), precipitation (b), temperature (c), specific humidity (d), cloud fraction (e), and PBLH (f) between 2020 and 2019 over the SCB and surrounding regions. All these meteorological parameters are from the GEOS-FP dataset. The vertical pressure velocity is prescribed at the PBLH and others are at the surface.

---

## Author Comment (AC5)

**Supplement of resonse letter #1**

**Section S1. Data filter criteria**

We applied a data quality control method used in Lu et al. (2020) to remove unreliable ozone data. Hourly observed data points were transformed into $Z$ scores, and then, the observed data were removed if the corresponding $Z_i$ met one of the following conditions: (1) $Z_i$ is larger or smaller than the previous one ($Z_{i-1}$) by 9 ($|Z_i - Z_{i-1}| > 9$), (2) The absolute value of $Z_i$ is greater than 4 ($|Z_i| > 4$), or (3) the ratio of the $Z$ value to the third-order center moving average is greater than 2 ($\frac{3Z_i}{Z_{i-1}+Z_i+Z_{i+1}} > 2$). The formula for calculating $Z_i$ is as follows:

$$Z_i = \frac{X_i - \bar{X}}{\sigma} \tag{1}$$

where $X_i$ represents the $i$-th item in the dataset, and $\bar{X}$ and $\sigma$ are the average and standard deviation of dataset $X$, respectively. The distribution of CNMEC sites over the SCB is shown in Figure 1 of the manuscript.

[Figure]

**Figure 1** Terrain elevations (a) and surface temperature and wind fields (b) over the SCB and surrounding regions. The spatial resolutions for (a) and (b) are 3 × 3 arc-minute and 0.25° × 0.25°, respectively. The white area in black line is Tibetan Plateau (with altitudes of 4–5 km a.s.l.), the yellow area in red line is the Yunnan-Kweichou Plateau (2–3 km a.s.l), the green area in circle is the SCB (0.5–1 km a.s.l).

[Figure]

**Figure 2** May-June mean differences in vertical pressure velocity (a), precipitation (b), temperature (c), specific humidity (d), cloud fraction (e), and PBLH (f) between 2020 and 2019 over the SCB and surrounding regions. All these meteorological parameters are from the GEOS-FP dataset. The vertical pressure velocity is prescribed at the PBLH and others are at the surface.

---

## Author Response (AR1)

**Point-by-point response letter**

Note: This file includes comments from the two referees and prof. Heini Wernli, the corresponding point-by-point responses, and the related changes in the manuscript. The black font are comments from the referees, and the red font are authors' responses as well as the related change clarifications.

**(1) Detailed response to comments from referee #1:**

The authors use high resolution nested-grid GEOS-Chem simulation, the eXtreme Gradient Boosting (XGBoost) machine learning method and the exposure−response relationship to determine the drivers and evaluate the health risks of the surface $O_3$ enhancements over the Sichuan basin (SCB) in May-June 2020, which are in contrast to an overall reduction in surface $O_3$ level across China. The authors first use the XGBoost machine learning method to correct the GEOS-Chem model-to-measurement $O_3$ discrepancy over the SCB, where large discrepancies between measured and modelled surface $O_3$ are found due to the complex terrain. The relative contributions of meteorology and anthropogenic emissions changes to the unexpected surface $O_3$ enhancements are then quantified with the combination of GEOS-Chem and XGBoost models. In order to assess the health risks caused by the unexpected $O_3$ enhancements over the SCB, total premature death mortalities are estimated.

The paper concluded that the unexpected changes in meteorology combined with the complex basin effect enhance downward transport of $O_3$ from upper troposphere, enhance biogenic emissions of volatile organic compounds (VOCs) and nitrogen oxides (NOx), speed up $O_3$ chemical production, and inhabit the ventilation of $O_3$ and its precursors, and therefore account for the surface $O_3$ enhancements over the SCB in May-June 2020. The total premature mortality due to the unexpected surface $O_3$ enhancements over the SCB has increased by 89.8% in May-June 2020 vs. 2019.

With a thoroughly review of this study, I would like to classify it as a very interesting and creative study. It is well written, structured, and its topic fits well in the scope of ACP. I believe that the results of this study could be of interest to the general atmospheric science community and should be in the literature. I recommend for publication after minor revisions.

**Response:** All your comments listed below have been addressed. Please check the point by point response as follows.

**General comments:**

**Comment [1-1]:** The authors use the XGBoost machine learning method to correct the GEOS-Chem model-to-measurement $O_3$ discrepancy over the SCB and then use the discrepancy corrected model to quantify the relative contributions of meteorology and anthropogenic emissions changes to the unexpected surface $O_3$ enhancements. This is a nice concept and I like it. However, this method used in present work can only separate the total meteorology or anthropogenic driven influences. For each individual meteorological or anthropogenic influence, the analysis is qualitative. As a result, I would suggest the authors to consolidate the analysis for the influence of each individual meteorological or anthropogenic factor. For example, as the community comments from Dr. Heini Wernli mentioned, the differences are on the order of 0.1 PVU (1 potential vorticity unit = $10^{-6}$ K $m^{-2}$ $kg^{-1}$ $s^{-1}$) for PV, which is very small, how the authors conclude from Fig. 6a that "the meteorology-induced surface ozone increase is mainly attributed to significant increases in temperature and downward potential vorticity" (p. 14 line 4). In addition, there are still some grammatical errors need to be corrected. I list part of them as bellow. I hope one of the authors with good command of English can go through the manuscript in detail or the ACP copy-editing service at the publication stage can help to correct all the glitches.

**Response:** In the revised version, we have double checked the analysis for the influence of each individual meteorological and anthropogenic factor. We have followed the suggestions of prof. Heini Wernli and removed the analysis for the potential vorticity. As a result, all concerns arise from the PV discussions are gone. Since we only performed very few analysis for the PV in the study, all revisions are minor. Instead, we have compared and analyzed the difference in vertical transport velocity at the PBLH between 2020 and 2019. We concluded that there is no strong evidence for the change in the horizontal transport from other regions (Figure 5(b)) and the vertical transport from the free troposphere to the surface (Figure 6 (a)) over the SCB in May-June 2020 vs. 2019 (Lefohn et al., 2012; Škerlak et al., 2014; Stohl et al., 2003; Wirth and Egger, 1999; Wang et al., 2019; Wang et al., 2020). In addition, we have corrected all grammatical errors listed below and one of the authors with good command of English have gone through the manuscript in detail to address the rest errors. Please check page 11, line 28-31 in the marked up file for details.

[Figure]

**Figure 5** Terrain elevations (a) and surface temperature and wind fields (b) over the SCB and surrounding regions. The spatial resolutions for (a) and (b) are 3 × 3 arc-minute and 0.25° × 0.25°, respectively. The white area in black line is Tibetan Plateau (with altitudes of 4–5 km a.s.l.), the yellow area in red line is the Yunnan-Kweichou Plateau (2–3 km a.s.l), the green area in circle is the SCB (0.5–1

km a.s.l).

[Figure]

**Figure 6** May-June mean differences in vertical pressure velocity (a), precipitation (b), temperature (c), specific humidity (d), cloud fraction (e), and PBLH (f) between 2020 and 2019 over the SCB and surrounding regions. All these meteorological parameters are from the GEOS-FP dataset. The vertical pressure velocity is prescribed at the PBLH and others are at the surface.

**Detailed comments:**

**Comment [1-2]:** Page 2, line 23, "Depending which …" should be "Depending on which …".

**Response:** Done. Please check page 2, line 23 in the marked up file for details.

**Comment [1-3]:** Page 3, line 15, "be applicable ..." should be "be applied...".

**Response:** Done. Please check page 3, line 15 in the marked up file for details.

**Comment [1-4]:** Page 3, line 19, "model mechanism..." should be "model mechanisms...".

**Response:** Done. Please check page 3, line 19 in the marked up file for details.

**Comment [1-5]:** Page 3, line 19, "discrepancy..." should be "a discrepancy...".

**Response:** Done. Please check page 3, line 19 in the marked up file for details.

**Comment [1-6]:** Page 3, line 38, "fourth largest..." should be "fourth-largest...".

**Response:** Done. Please check page 3, line 38 in the marked up file for details.

**Comment [1-7]:** Page 4, line 4, "highly industrialized region..." should be "highly industrialized regions...".

**Response:** Done. Please check page 4, line 4 in the marked up file for details.

**Comment [1-8]:** Page 4, line 16, "After removing unreliable measurements with the filter criteria used in Lu et al. (2020)". Please add the data filter criteria to the supplement.

**Response:** We have included the data filter criteria to the supplement. Please check section S1 for details.

**Comment [1-9]:** Page 4, line 41, "3-hour interval..." should be "3-hour intervals...".

**Response:** Done. Please check page 4, line 41 in the marked up file for details.

**Comment [1-10]:** Page 5, equation (1) should be divided into equations (1), (2), (3).

**Response:** Done. Please check page 5, line 25-27 in the marked up file for details.

**Comment [1-11]:** Page 6, line 14, "a10-fold" should be "a 10-fold".

**Response:** Done. Please check page 6, line 17 in the marked up file for details.

**Comment [1-12]:** Page 6 equations (4) and (5), the definitions of XG_Emis and XG_Met are missing.

**Response:** We have included the statement "where the acronyms are similar to those in equations (4) and (5) but for GEOS-Chem-XGBoost method" in the revised version. Please check page 7, line 7 in the marked up file for details.

**Comment [1-13]:** Page 8, line 32, "relative poorer..." should be "relatively poorer...".

**Response:** Done. Please check page 8, line 42 in the marked up file for details.

**Comment [1-14]:** Page 9, line 1, "each individual model…" should delete "individual".
**Response:** Done. Please check page 9, line 11 in the marked up file for details.

**Comment [1-15]:** Page 9, line 1, "which offer…" should be "which offers".
**Response:** In the revised version, should be "offer". Please check page 9, line 11 in the marked up file for details.

**Comment [1-16]:** Page 10, line 19, "slightly with…" should be "slightly from…".
**Response:** Done. Please check page 10, line 27 in the marked up file for details.

**Comment [1-17]:** Page 12, line 38, "Table S4 and S5" should be "Tables S4 and S5".
**Response:** Done. Please check page 13, line 12 in the marked up file for details.

**Comment [1-18]:** Page 13, line 20, "largest populations" should be "the largest populations".
**Response:** Done. Please check page 13, line 36 in the marked up file for details.

**Comment [1-19]:** Page 14, line 13, "in May-June 2020" should be "during May-June in 2020".
**Response:** Done. Please check page 14, line 28 in the marked up file for details.

**Comment [1-20]:** Figures 1, 6, 7 should add the corresponding latitude and longitude.
**Response:** We have included latitude and longitude information in Figures 1, 6, and 7 . Please check the marked up file for details.

**(2) Detailed response to comments from referee #2:**

This manuscript described an assessment of relative influence of meteorology and emissions on the surface $O_3$ enhancements over the Sichuan basin (SCB) in May-June 2020 using high resolution nested-grid GEOS-Chem simulation and the eXtreme Gradient Boosting (XGBoost) machine learning model. Furthermore, the health risks of the surface $O_3$ enhancements in terms of various premature mortalities are also evaluated by using the exposure−response relationship. The surface $O_3$ enhancements over the SCB are in contrast to an overall reduction in surface $O_3$ level across China. The authors first demonstrated the effectiveness of XGBoost to mitigate the model prediction discrepancy over the complex terrain over the SCB. The relative contributions of meteorology and anthropogenic emissions changes to the unexpected

surface $O_3$ enhancements are then quantified with the combination of GEOS-Chem and XGBoost models. The authors concluded that the unexpected surface $O_3$ enhancements over the SCB is attributed to the unexpected changes in meteorology combined with the complex basin effect, which caused an increase in the total premature mortality of 89.8% in May-June 2020 vs. 2019. In general, the topic is interesting and the majority of the works are sound. It is well organized, written and analyzed convincingly, and its topic fits well in the scope of ACP. I recommend for publication after addressing the followings comments.

**Response:** All your comments listed below have been addressed. Please check the point by point response as follows.

**General comments:**

**Comment [2-1]:** The assessment of the influence of meteorology and emissions is based on the premise that the GEOS-Chem-XGBoost effectively corrected the model discrepancy over the SCB for May-June 2020. Since the ozone formation is highly non-linear and has strong dependence on its precursor levels and meteorology, the training data should cover the variation range, at least, of the key ozone precursors or meteorology. The training and validation of the XGBoost with observations for a specific period may not be applicable for all conditions especially for the case that significant emissions or meteorology changes occurred. In this study, the authors use a full seasonal cycle of hourly measurements in 2019 at each site over SCB as the learning samples, and GEOS-Chem input of emissions and meteorological parameters, output concentrations of atmospheric constituents, and time information as training input data. The usage of the GEOS-Chem-XGBoost is valid only if the range of variations for the training data in 2019 cover that in May-June 2020. So a few discussion or clarification is needed to consolidate the usage of this method.

**Response:** In the revised version, we have included the probability density functions (PDF) of the key ozone precursors (i.e., $NO_2$, CO and HCHO) and meteorological parameters (planetary boundary height layer (PBLH), temperature, specific humidity) in the two most densely populated cities over the SCB (i.e., Chengdu and Chongqing). We verified that the training data (a full seasonal cycle of 2019) cover the variation ranges of $O_3$ precursors in May-June 2020 (Figures S2 and S3). Please check page 6, line 37-40 in the marked up file for details.

[Figure]

**Figure S2.** Probability density functions (PDFs) of hourly planetary boundary layer height (PBLH), temperature at 2 m, and relative humidity in the whole 2019 (blue) and May-June 2020 (red) at Chengdu and Chongqing cities over the SCB, from the GEOS-FP meteorology fields that are used to drive the GEOS-Chem model and to train/predict the ozone bias. We group each data to 10 bins, and frequencies are calculated for each bin.

[Figure]

**Figure S3.** Same as Fig.S2, but for hourly concentrations of GEOS-Chem $NO_2$, CO, and HCHO. The difference between 2019 and 2020 in GEOS-Chem only reflects meteorological effects.

**Comment [2-2]:** The aggregate meteorological influence and the aggregate

anthropogenic influence are quantified with the GEOS-Chem-XGBoost method. However, the analysis for the influence of each individual meteorological or anthropogenic factor, based on the differences between 2020 and 2019 over the SCB and surrounding regions, are qualitative. The analysis for the potential vorticity (PV) is needed to be verified and modified. The differences in PVU over the SCB and surrounding regions are very small between 2020 and 2019. As a result, the meteorology-induced surface ozone increase over SCB may be attributed to other meteorological anomalies rather than PV. Normally, the influence of stratospheric intrusions on near-surface ozone can be evident on a specific short period but can't last for months. I recommend the authors to modify the PV associated deduction and temper the description for its influence.

**Response:** In the revised version, we have double checked the analysis for the influence of each individual meteorological and anthropogenic factor. We have followed the suggestions of prof. Heini Wernli and removed the analysis for the potential vorticity. As a result, all concerns arise from the PV discussions are gone. Since we only performed very few analysis for the PV in the study, all revisions are minor. Instead, we have compared and analyzed the difference in vertical transport velocity at the PBLH between 2020 and 2019. We concluded that there is no strong evidence for the change in the horizontal transport from other regions (Figure 5(b)) and the vertical transport from the free troposphere to the surface (Figure 6 (a)) over the SCB in May-June 2020 vs. 2019 (Lefohn et al., 2012; Škerlak et al., 2014; Stohl et al., 2003; Wirth and Egger, 1999; Wang et al., 2019; Wang et al., 2020). Please check page 11, line 28-31 in the marked up file for details.

[Figure]

**Figure 5** Terrain elevations (a) and surface temperature and wind fields (b) over the SCB and surrounding regions. The spatial resolutions for (a) and (b) are 3 × 3 arc-minute and 0.25 ° × 0.25 °, respectively. The white area in black line is Tibetan Plateau (with altitudes of 4–5 km a.s.l.), the yellow area in red line is the Yunnan-Kweichou Plateau (2–3 km a.s.l), the green area in circle is the SCB (0.5–1

km a.s.l).

[Figure]

**Figure 6** May-June mean differences in vertical pressure velocity (a), precipitation (b), temperature (c), specific humidity (d), cloud fraction (e), and PBLH (f) between 2020 and 2019 over the SCB and surrounding regions. All these meteorological parameters are from the GEOS-FP dataset. The vertical pressure velocity is prescribed at the PBLH and others are at the surface.

**Comment [2-3]:** There are still some grammatical errors which needs further careful check. For example, the usage of "emission" and "emissions" is sometimes misleading. Referee #1 has listed part of them.

**Response:** In the revised version, we have corrected all grammatical errors listed

below and one of the authors with good command of English have gone through the manuscript in detail to address the rest errors. Please check the marked up file for details.

**Specific corrections:**

**Comment [2-4]:** Page 6 line 4, "All these training input data are summarized in Table S1 and have been standardized". Please describe in more details for "standardized".

**Response:** In the revised version, we have included detailed description of " standardized" in section S2 in the supplement. Please check the marked up file for details. As the parameters listed in Table S1 are different in units and magnitudes, which could lead to unstable performance of the training model. Therefore, we standardized all the parameters before using them for model training. The standardized process is expressed as below:

$$D_i = \frac{P_i - \mu}{\sigma} \tag{1}$$

where $P_i$, $\mu$, and $\sigma$ are the $i$-th parameter, the average, and the standard deviation of the training input dataset listed in Table S1, respectively. D$i$ represents the standardized value used for model training.

**Comment [2-5]:** It would be helpful if Figure 1a-c showed the 1std of mean value.

**Response:** Done. Please check Figure 1a-c in the marked up file for details.

**Comment [2-6]:** In Figure 4 (a), the ozone variability is smoother than that in Figure 3 (a). I wonder if a certain running average is used. Please clarify.

**Response:** In the revised version, we have stated that values shown in this figure are 7-day running average. Please check the caption of Figure 4 in the marked up file for details.

**Comment [2-7]:** Figures 1, 6, 7 should add the corresponding latitude and longitude.

**Response:** We have included latitude and longitude information in Figures 1, 6, and 7 . Please check the marked up file for details.

**Comment [2-8]:** In Figure 8, the titles of each subplot are not needed.

**Response:** We have removed the titles of each subplot. Please check Figure 8 in the marked up file for details.

**Comment [2-9]:** Can the authors show the differences in vertical transport velocity at the PBLH between 2020 and 2019? This could help the reader to understand the vertical transport at the studies period.

**Response:** In the revised version, we have showed the differences in vertical transport velocity at the PBLH between 2020 and 2019 over the SCB, and the resulting discussion are presented accordingly. Please check page 11, line 28-31 in the marked up file for details.

**Comment [2-10]:** Please make sure all references follow the ACP format.
**Response:** Done. Please check the references in the marked up file for details.

**(3) Detailed response to comments from prof. Heini Wernli:**

The way how potential vorticity (PV) is used in this study as a tracer of stratospheric air is very confusing and most likely flawed. Many previous studies used potential vorticity changes along air parcel trajectories to identify events of stratosphere-to-troposphere transport, which can significantly affect near-surface ozone concentrations (e.g., Wirth and Egger, 1999; Stohl et al., 2003; Lefohn et al., 2012; Škerlak et al., 2014). These and many other studies clearly show that it is important to consider the vertical structure of the PV field and the evolution of PV along trajectories associated with observations of enhanced surface ozone. However, in the study by Sun et al. it is not clear at what level and how PV has been evaluated. The abstract mentions "downward potential vorticity" (p. 1 line 34), but PV is a scalar, it has no orientation, therefore "downward PV" does not make sense to me. Then an "increase in PV" is mentioned (p. 2 line 29), but it is not clear where this increase should occur. When stratosphere-to-troposphere transport down to surface occurs, then typically the originally high stratospheric PV values is reduced during the transport by diabatic processes and therefore the air parcel may arrive at the surface without high PV but still with elevated ozone concentrations. Near the page break of p. 10/11 it is mentioned that vertical convection "will continuously intensify the cyclonic vorticity over the Sichuan basin" – it is not clear to me whether this sentence refers to relative vorticity or to PV? And is it about the downward transport of stratospheric PV or about the diabatic production of low-level PV? The latter process would most likely not lead to enhanced surface ozone values. And finally, Fig. 6a shows "differences in PV … between 2020 and 2019". How were these PV fields calculated and at what level is PV considered here? The field shown in Fig. 6a is not useful when investigating a potential influence of stratospheric intrusions on near-surface ozone. The differences are on the order of 0.1 PVU (1 potential vorticity unit = 10-6 K m-2 kg-1 s-1), which is very small. I don't understand how the authors conclude from Fig. 6a that "the meteorology-induced surface ozone increase is mainly attributed to significant increases in temperature and downward potential vorticity" (p. 14 line 4).

References:

Lefohn, A. S., H. Wernli, D. Shadwick, S. J. Oltmans, and M. Shapiro, 2012: Quantifying the importance of stratospheric-tropospheric transport on surface ozone concentrations at high- and low-elevation monitoring sites in the United States. Atmos. Environ., 62, 646–656.

Škerlak, B., M. Sprenger, and H. Wernli, 2014: A global climatology of stratosphere-troposphere exchange using the ERA-Interim data set from 1979 to 2011. Atmos. Chem. Phys., 14, 913–937.

Stohl, A., H. Wernli, M. Bourqui, C. Forster, P. James, M. A. Liniger, P. Seibert, and M. Sprenger, 2003: A new perspective of stratosphere-troposphere exchange. Bull. Amer. Meteor. Soc., 84, 1565-1573.

Wirth, V., and J. Egger, 1999: Diagnosing extratropical synoptic-scale stratosphere–troposphere exchange: A case study. Quart. J. Roy. Meteor. Soc., 125, 635–655.

**Response:** Thanks very much for your comments and your detailed explanations in the email (not shown in the community comments) regarding the usage of potential vorticity (PV), which help us to avoid a controversial deduction in the revision.

We have quantified the aggregate meteorological influence and the aggregate anthropogenic influence on the unexpected surface ozone enhancements over the SCB with the GEOS-Chem-XGBoost method. It is no doubt that the unexpected surface ozone enhancements over the SCB is induced by meteorology anomaly. However, the GEOS-Chem-XGBoost method cannot separate the influence of each individual meteorological or anthropogenic factor. As a result, we compare the differences in many meteorological or anthropogenic factors between 2020 and 2019 over the SCB and surrounding regions to probe qualitatively each individual influence. In previous version, we attempted to use the potential vorticity (PV) at the planetary boundary layer height (PBLH) as a tracer to evaluate the stratosphere-to-troposphere transport. We have very little sense regarding how much difference in PVU at the PBLH can be called large or small, which resulted in a controversial deduction. After reading the references that you listed at the end of the comments as well as your explanations in the email, we give up to use the PV at the PBLH as a tracer to evaluate the stratosphere-to-troposphere transport and removed all PV related content and discussions in the revised version. As a result, all your concerns mentioned above are gone. Since we only performed very few analysis for the PV in the study, all revisions are minor.

Instead, we have compared and analyzed the difference in vertical transport velocity at the PBLH between 2020 and 2019. We concluded that there is no strong evidence for the change in the horizontal transport from other regions (Figure 5(b)) and the vertical transport from the free troposphere to the surface (Figure 6 (a)) over the SCB

in May-June 2020 vs. 2019 (Lefohn et al., 2012; Škerlak et al., 2014; Stohl et al., 2003; Wirth and Egger, 1999; Wang et al., 2019; Wang et al., 2020). Please check page 11, line 28-31 in the marked up file for details.

[Figure]

**Figure 5** Terrain elevations (a) and surface temperature and wind fields (b) over the

SCB and surrounding regions. The spatial resolutions for (a) and (b) are 3 × 3 arc-minute and 0.25 ° × 0.25 °, respectively. The white area in black line is Tibetan Plateau (with altitudes of 4–5 km a.s.l.), the yellow area in red line is the Yunnan-Kweichou Plateau (2–3 km a.s.l), the green area in circle is the SCB (0.5–1 km a.s.l).

[Figure]

**Figure 6** May-June mean differences in vertical pressure velocity (a), precipitation (b), temperature (c), specific humidity (d), cloud fraction (e), and PBLH (f) between 2020 and 2019 over the SCB and surrounding regions. All these meteorological parameters are from the GEOS-FP dataset. The vertical pressure velocity is prescribed at the PBLH and others are at the surface.